# General health of patients with diabetic macular edema—The LIPSIA study

**Catharina Busch**[1]*, **Julius L. Katzmann**[2], **Claudia Jochmann**[1], **Jan Darius Unterlauft**[1], **Daniela Vollhardt**[1], **Peter Wiedemann**[1], **Ulrich Laufs**[2]☯, **Matus Rehak**[1]☯

1 Department of Ophthalmology, University Hospital Leipzig, Leipzig, Germany, 2 Department of Cardiology, University Hospital Leipzig, Leipzig, Germany

☯ These authors contributed equally to this work.
* busch.catharina@gmail.com

## Abstract

### Purpose

Cardiovascular risk factors such as hypertension or dyslipidemia can influence the incidence and progression of diabetic retinopathy (DR) and diabetic macular edema (DME). The aim of this study is to describe the comorbidities in patients with DME.

### Methods

Prospective, monocentric observational study. Patients presenting for the treatment of DME received laboratory and clinical examinations including 24-hour blood pressure measurement.

### Results

Seventy-five consecutive patients were included in the study. The mean age was 61.0 ± 14.5 years, and 83% had type 2 diabetes. The mean body mass index (BMI) was 32.8 ± 6.0 kg/m². Overweight (BMI $\geq$ 25 kg/m²) was present in 92% of all patients. HbA1c values were > 7.0% in 57%. Although 87% of the patients already received antihypertensive therapy, the blood pressure (BP) of 82% was still above the recommended target values of systolic < 140 mmHg and diastolic < 80 mmHg. An insufficient nocturnal fall of the systolic BP (< 10%, non-dipping or reverse dipping) was observed in 62%. In 83% of the patients the glomerular filtration rate was $\leq$ 90 ml/min/1.73m². Despite 65% of the cohort already receiving lipid-lowering therapy, LDL cholesterol was above the target value of 1.4 mmol/l in 93%. All patients had at least one cardiovascular risk factor in addition to diabetes (overweight, hypertension, insufficient nocturnal BP fall, dyslipidemia, or renal dysfunction) and 86% had $\geq$ 3 risk factors.

### Conclusion

DME patients are characterized by highly prevalent cardiovascular risk factors that are poorly controlled. These comorbidities reduce the prognosis and negatively influence existing DR and DME. The data reveal an important opportunity for improving patient care by

**Data Availability Statement:** All relevant data are within the manuscript.

**Funding:** The study was funded by an grant to MR by Novartis Pharma (Grant number:

MRTH258A_FVMR002). The funders had no role in study design, data collection and analysis, decision to publish, or preparation of the manuscript.

**Competing interests:** The study was funded by a grant to MR by Novartis Pharma (Grant number: MRTH258A_FVMR002). This does not alter our adherence to PLOS ONE policies on sharing data and materials.

interaction of the ophthalmologist with the general practitioner and internal specialists for the detection and treatment of these conditions.

## Introduction

The number of people with diabetes worldwide is estimated to increase from 451 million in 2017 to 693 million in 2045 [1]. Diabetic retinopathy (DR), and especially diabetic macular edema (DME), is the leading cause of significant vision loss in patients of working age [2]. Pan-retinal photocoagulation and intravitreal treatment with vascular endothelial growth factor (VEGF) inhibitors can often prevent or at least delay the progression of DR into severe proliferative forms [3–5]. The functional long-term outcome of patients with DME has also been significantly improved by the introduction of intravitreal therapy with VEGF inhibitors or steroids [6]. However, despite intensive therapy, the desired treatment success is not always achieved, which could be due to systemic factors. About 60–70% of diabetic patients have at least one comorbidity and 25–30% have more than 2 comorbidities when diagnosed with diabetes [7]. The most frequent comorbidities are hypertension, overweight/obesity, and dyslipidemia [7]. It is known that an elevated blood pressure in particular has a negative impact on diabetic retinopathy [8]. In addition, a negative influence of co-existing obesity and dyslipidemia is also discussed [9–11]. In daily clinical practice, however, ophthalmologists may not be fully aware of the patients' general health and thus of possible negative factors influencing DR.

Therefore, the aim of the study was to describe a cohort of patients with DME regarding existing comorbidities and potential risk factors for DR progression.

## Methods

This is a prospective, monocentric observational study. The study (EK 222/18) was approved by the ethics commission at the University of Leipzig and was conducted according to the Declaration of Helsinki guidelines. A written consent was obtained from all study participants.

### Study participants

Patients were included into the study from August 1, 2018 to December 31, 2019. The following inclusion criteria had to be fulfilled: (1) age $\geq$ 18 years; (2) diabetes mellitus type 1 or type 2; (3) treatment-naïve DME with intra- and/or subretinal fluid seen on Spectral Domain optical coherence tomography (SD-OCT), or DME recurrence with last anti-VEGF therapy $\geq$ 3 months ago, and/or last dexamethasone implant, intravitreal triamcinolone, or macular laser $\geq$ 6 months ago. Exclusion criteria were: (1) concomitant ocular disease that could cause macular edema (including choroidal neovascularization from any cause, retinal vein occlusion, vitreomacular traction, uveitis, and recent intraocular surgery); (2) any concomitant ocular or neurological condition that could affect vision except cataract; (3) pregnancy.

### Data collection

Demographic data as well as general medical and ophthalmological history were collected at study enrollment. In addition, a comprehensive clinical ophthalmological examination and imaging (OCT, wide-field fundus picture, wide-field fluorescein angiography, OCT angiography), an ambulatory 24-hour blood pressure measurement and a comprehensive laboratory testing (fasting blood test) were performed.

The main outcome was baseline characteristics regarding the general health of our cohort. The demographic and clinical characteristics were evaluated using traditional descriptive methods. Statistical analysis was performed with SPSS Statistics 26 (IBM, Armonk, NY, USA).

## Results

Seventy-five patients were included in the study. The mean age was 61.0 ± 14.5 years and 57% were male (43/75 patients). The mean body mass index (BMI) was 32.8 ± 6.0 kg/m² (Table 1). A BMI ≥ 25 kg/m² (overweight) and ≥ 30 kg/m² (obesity) was present in 92% (60/65 patients) and 69% (45/65 patients) of patients, respectively. The mean diabetes duration was 19.1 ± 13.2 years. Most of the patients had type 2 diabetes (83%, 62/75 patients). Other vascular complications due to diabetes, such as coronary heart disease or peripheral artery disease, were already known in almost half of the patients at the beginning of the study (49%, 34/69 patients). Further details on comorbidities and medications are presented in Table 1.

**Table 1. Descriptive characteristics, n = 75.**

| | |
|---|---|
| Age, years, mean (SD) | 61.0 (14.5) |
| Male, n (%) | 43 (57.3) |
| Body-Mass Index (BMI), kg/m², mean (SD) | 32.8 (6.0) |
| BMI ≥ 25 kg/m², n (%) | 60/65 (92.3) |
| BMI ≥ 30 kg/m², n (%) | 45/65 (69.2) |
| BMI ≥ 35 kg/m², n (%) | 26/65 (40.0) |
| BMI ≥ 40 kg/m², n (%) | 7/65 (10.8) |
| Type of diabetes, n (%) | |
| Type-1 Diabetes | 13 (17.3) |
| Type-2 Diabetes | 62 (82.7) |
| Duration of diabetes, years, mean (SD), n = 73 | 19.1 (13.2) |
| Smoking status, n (%), n = 71 | |
| Current smoker | 9 (12.7) |
| Former smoker | 34 (47.9) |
| Non-smoker | 28 (39.4) |
| Known hypertension, n (%) | 67/71 (89.3) |
| Known vascular complications, n (%) | |
| Status post stroke | 8/69 (11.6) |
| Coronary heart disease | 22/69 (31.9) |
| Peripheral artery disease | 10/69 (14.5) |
| Diabetic nephropathy | 18/69 (26.1) |
| Diabetic neuropathy | 17/69 (24.6) |
| None | 35/69 (50.7) |
| Pharmacological therapies | |
| Insulin, n (%) | 62 (81.6) |
| Only oral antidiabetics, n (%) | 12 (16.0) |
| Only diet | 1 (1.3) |
| Antihypertensive treatment, n (%) | 65 (86.7) |
| Number of antihypertensive agents, mean (SD), n = 65 | 3.2 (1.6) |
| Statins, n (%) | 49/72 (65.3) |

SD—standard deviation.

Approximately half of the patients were naïve regarding DME therapy (36/75 patients, 48%). Non-naïve eyes were mostly pretreated with anti-VEGF injections (37/39 patients, 95%). The mean duration since last DME treatment was 15.6 ± 21.1 months. The ophthalmological baseline characteristics are shown in Table 2.

**Table 2. Ophthalmological characteristics.**

| | |
|---|---|
| Treatment-naïve DME, n (%) | 36 (48.0) |
| Prior anti-VEGF treatment, n (%) | 37 (49.3) |
| No. of prior anti-VEGF injections, mean (SD) | 8.8 (6.1) |
| Prior therapy with DEX implant, n (%) | 5 (6.7) |
| Prior therapy with FA implant, n (%) | 2 (2.7) |
| Prior therapy with IVTA, n (%) | 1 (1.3) |
| Duration since last intravitreal injection, months, mean (SD) | 15.6 (21.1) |
| Prior macular laser, n (%) | 8 (10.7) |
| DR severity, n (%), n = 74 | |
| None | 3 (4.0) |
| Mild NPDR | 27 (36.0) |
| Moderate NPDR | 15 (20.0) |
| Severe NPDR | 17 (23.0) |
| PDR | 12 (16.0) |
| Prior panretinal photocoagulation, n (%) | 25 (33.3) |
| Lens status, n (%) | |
| Phakic | 51 (68.0) |
| Pseudophakic | 24 (32.0) |
| Best corrected visual acuity, letters, mean (SD) | 69.8 (11.4) |
| Central subfield thickness, μm, mean (SD), n = 74 | 416.7 (114.0) |

DEX—dexamethasone, DME—diabetic macular edema, FA—fluocinolone acetonide, IVTA—intravitreal triamcinolone, NPDR—non-proliferative diabetic retinopathy, PDR—proliferative diabetic retinopathy, SD—standard deviation, VEGF—vascular endothelial growth factor.

Although 87% of the patients (65/75 patients) were already on antihypertensive medication at study enrollment, 82% (55/67 patients) had blood pressure values in the ambulatory blood pressure measurement that were above the values recommended for diabetic patients [12] (systolic ≥ 140 mmHg and/or diastolic ≥ 80 mmHg, Table 3). Maximum systolic values > 200 mmHg were observed in 48% of the patients (32/67 patients). An increased mean diastolic value (≥ 80 mmHg) was found in 57% of the patients (38/67 patients). Only 37% (23/62 patients) showed a sufficient nocturnal BP fall (dipping, ≥ 10% of the daily systolic average value). A non-dipping (nocturnal reduction < 10%) was observed in 31% (19/62 patients) and a reverse dipping (nocturnal mean systolic value above the daytime average) in 32% (20/62 patients). An evaluation of the dipping pattern was not possible for five patients because of artefacts during nighttime measurements.

**Table 3. Baseline blood pressure values, n = 67.**

| | Minimum | Maximum | Mean |
|---|---|---|---|
| Systolic, mmHg, mean (SD) | 121.1 (22.6) | 196.4 (22.5) | 158.2 (22.9) |
| Diastolic, mmHg, mean (SD) | 56.5 (10.2) | 115.2 (20.2) | 81.1 (10.3) |
| MAP, mmHg, mean (SD) | 80.1 (12.2) | 138.4 (16.9) | 106.9 (12.3) |
| Heart rate, bpm, mean (SD) | 63.2 (10.4) | 95.8 (16.2) | 74.5 (11.1) |
| Pulse pressure, mmHg, mean (SD) | 46.3 (20.8) | 106.2 (23.9) | 77.6 (20.6) |

bpm—beats per minute, MAP—mean arterial pressure, SD—standard deviation.

**Table 4. Laboratory parameters, n = 72.**

| | Mean (SD) | Reference range | |
|---|---|---|---|
| HbA1c, %, n = 70 | 7.3 (1.3) | 5.0–12.2 | > 7.0% in 57% (40/70 patients) |
| | | | > 8.0% in 23% (16/70 patients) |
| Fasting glucose, mmol/l | 9.6 (3.2) | 4.1–5.6 | increased in 93% (67/72 patients) |
| Creatinine, µmol/l | 98.6 (56.3) | 59–104 | increased in 28% (20/72 patients) |
| Cystatin C, mg/l | 1.39 (0.68) | 0.61–0.95 | increased in 83% (60/72 patients) |
| GFR* ml/min/1.73m², n = 71 | 68.3 (27.9) | > 90 | ≤ 90 ml/min/1.73 m² in 83% (59/71 patients) |
| | | | ≤ 60 ml/min/1.73 m² in 39% (28/71 patients) |
| | | | ≤ 45 ml/min/1.73 m² in 23% (16/71 patients) |
| | | | ≤ 30 ml/min/1.73 m² in 10% (7/71 patients) |
| Triglycerides, mmol/l | 1.76 (1.07) | < 1.7 | increased in 38% (27/72 patients) |
| HDL cholesterol, mmol/l | 1.41 (0.41) | > 1.03 | decreased in 15% (11/72 patients) |
| LDL cholesterol, mmol/l | 2.54 (0.91) | < 4.2 | ≥ 1.4 mmol/l in 93% (67/72 patients) |
| Lipoprotein(a), nmol/l | 73.7 (95.2) | < 20 | > 75 nmol/l in 28% (20/72 patients) |
| Troponin T**, pg/ml | 22.9 (20.6) | < 14 | increased in 51% (36/71 patients) |
| NT-ProBNP, pg/ml | 462.5 (902.7) | < 263 | increased in 31% (22/72 patients) |

*Creatinine and Cystain C based.

**high sensitivity. HbA1c—hemoglobin A1c, HDL—high density lipoprotein, GFR—glomerular filtration rate, LDL–low-density lipoprotein, NT-ProBNP—n-terminal pro brain natriuretic peptide, SD—standard deviation.

The laboratory results and existing comorbidities are presented in Table 4 and Fig 1, respectively. The HbA1c value was above 7.0% in 57% (40/70 patients) and above 8.0% in 23% of the patients (type 1 diabetes cohort: 31%, (4/13 patients) and type 2 diabetes cohort: 21% (12/57 patients)). Fasting glucose was above the recommended therapy target of 6.9 mmol/l in 78%

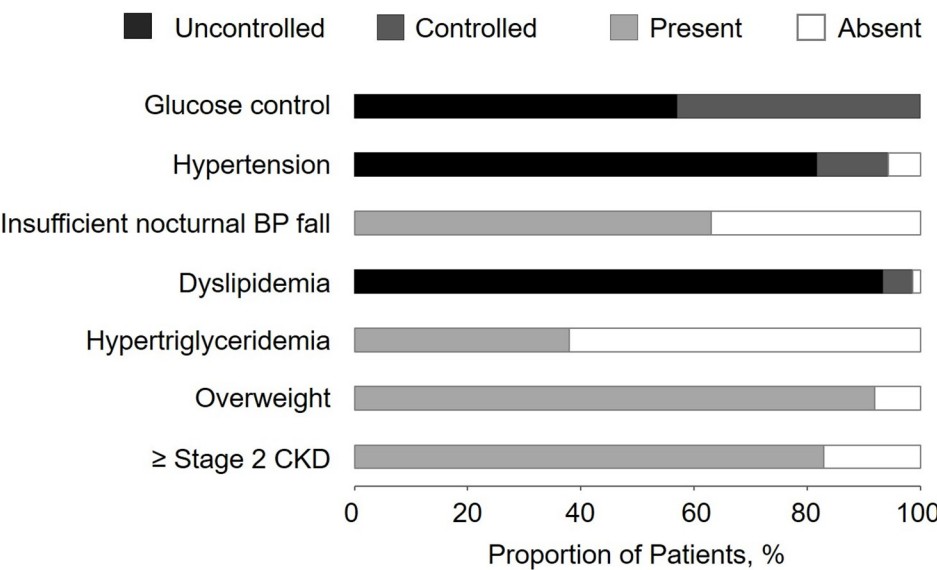

**Fig 1. Proportion of patients with dyslipidemia (low-density lipoprotein cholesterol ≥ 1.4 mmol/l), overweight (body mass index [BMI] ≥ 25 kg/m²), ≥ stage 2 chronic kidney disease (CKD, glomerular filtration rate [GFR] ≤ 90 ml/min/1.73 m²), hypertension (systolic blood pressure [BP] ≥ 140 mmHg and/or diastolic BP ≥ 80 mmHg), obesity (BMI ≥ 30 kg/m²), insufficient nocturnal PB fall (< 10% of the daily systolic average value), insufficient glucose control (HbA1c > 7.0%), ≥ stage 3 CKD (GFR ≤ 60 ml/min/1.73 m²) and hypertriglyceridemia (fasting triglycerides ≥ 1.7 mmol/l).**

(56/72 patients) [13]. In 83% of the cohort (60/72 patients) the glomerular filtration rate (GFR) was $\leq$ 90 ml/min/1.73m$^2$. Even though 68% of the patients received lipid-lowering therapy, 93% (67/72 patients) had an low-density lipoprotein (LDL) cholesterol above the recommended value of 1.4 mmol/l for DME patients [12]. Elevated triglyceride levels were found in 38% of the patients. Lipoprotein(a) (Lp(a)), an independent cardiovascular risk factor, was above 75 nmol/l in 28% of the patients (20/72 patients). Over half of the patients also had elevated troponin T levels (36/71 patients, 51%).

All patients had at least one of the five following comorbidities or cardiovascular risk factors: Overweight (BMI $\geq$ 25 kg/m$^2$), hypertension (systolic BP values $\geq$ 140 mmHg systolic and/or diastolic values $\geq$ 80 mmHg), pathological nocturnal blood pressure fall (non-dipping/reverse dipping), dyslipidemia (defined as LDL cholesterol $\geq$ 1.4 mmol/l) and/or fasting triglycerides $\geq$ 1.7 mmol/l, and renal insufficiency (GFR $\leq$ 60 ml/min/1.73m$^2$). Three or more comorbidities were found in 86% (56/65 patients), $\geq$ 4 comorbidities in 49% (32/65 patients), and all 5 comorbidities in 12% (8/65 patients).

## Discussion

This study describes the cardiovascular risk factors of a real-world cohort treated for diabetic macular edema at a German university hospital and reveals highly prevalent cardiovascular risk factors that are poorly controlled in a very high number of patients. Most patients were overweight (92%), had insufficient blood pressure control (82%), dyslipidemia (93%), and impairment of renal function (83%).

In daily clinical practice, patients regularly experience a progression of DR despite intensive ophthalmological treatment. The ETDRS (Early Treatment Diabetic Retinopathy Study) showed a DR progression rate of up to 28% despite panretinal laser treatment in PDR patients [3]. Systemic factors must be considered as the cause of this progression. Previous studies have already revealed that systemic factors, in particular, HbA1c, blood pressure, overweight, and dyslipidemia influence DR.

The optimal HbA1c value is determined individually for each patient and depends on age, life expectancy, the occurrence of severe hypoglycemia, and the presence of vascular comorbidities [14]. The American College of Physicians recommends a HbA1c value of 7–8% for most type 2 diabetics [14], while the European Society of Cardiology recommends to target a near-normal HbA1c value < 7% to decrease the risk for microvascular complications [12]. In our study, 57% of patients showed an HbA1c above 7.0% and 23% of the patients were above 8.0%. Previous studies have already shown that an increased HbA1c value is associated with an increased risk of DR and DME development and DR progression [10, 15–18]. The probability of DR or DME development and progression can be influenced positively by lowering the HbA1c level [15].

Another important factor influencing DR is hypertension. In our study, 82% of the patients showed an inadequate blood pressure control. The Wisconsin Epidemologic Study in type 1 diabetics provided clear evidence that an elevated systolic and/or diastolic blood pressure has a negative impact on DR incidence and progression and increases the risk of DME [15]. Present hypertension at baseline increased the probability of DR or DME progression by 91% and 40%, respectively, within the study period of 14 years [15]. However, not only the initial blood pressure value but also the change in blood pressure values during the follow-up had an influence. A 10 mmHg increase in diastolic blood pressure within the first four years of the study increased the probability of DR progression by 35% [15] and reduced the probability of a significant DR improvement by 22% [16]. An increased incidence of DR was also found among type 2 diabetics with increasing systolic and diastolic blood pressure values [10, 19, 20]. The

extent to which an anti-hypertensive therapy influences the severity of DR remains to be discussed. A Cochrane review of 15 randomized controlled trials in type 1 and type 2 diabetics showed a positive effect of anti-hypertensive therapy on the 4- to 5-year incidence of DR [21]. However, a clear effect on the rate of progression of existing DR or on the incidence of DME could not be established [21]. A post-hoc analysis of sham-treated eyes in the MEAD study revealed a significantly better 3-year result in eyes without prior hypertension (difference: +5.8 letters). Furthermore, anatomically better results were found in eyes with anti-hypertensive therapy [22], suggesting a positive effect of anti-hypertensive treatment on edema reduction.

Due to the higher predictive power regarding the cardiovascular risk, ambulatory 24-hour blood pressure measurements should be preferred to single measurements and initiated in every patient with DR [23]. A normal circadian blood pressure variability is characterized by higher blood pressure values during daytime and a decrease of 10–20% during nighttime (dipping) [23]. Patients with non-dipping (nocturnal systolic reduction of less than 10%) and reverse dipping (nighttime systolic blood pressure values higher than daytime values) have a significantly higher risk of cardiovascular events such as myocardial infarction or stroke, as well as higher cardiovascular mortality and total mortality [24]. A non-dipping or reverse dipping was observed in 63% of our cohort. It is known that the nighttime blood pressure can predict cardiovascular events better than daytime values [23]. In addition, nocturnal blood pressure behavior also seems to correlate better with the severity of DR. Klein et al. showed in a normotensive cohort of type 1 diabetics that systolic nocturnal blood pressure values and the systolic night-to-day ratio correlated significantly with DR severity [25]. Cardoso et al. reported in a longitudinal study in type 2 diabetics significantly higher systolic and diastolic daytime and nighttime values, as well as a reduced nocturnal fall at study baseline in patients with consecutive DR development or progression during the course of the study [10]. Cardoso et al. also confirmed in their study that an ambulatory 24-hour blood pressure monitoring has a higher predictive power regarding a possible DR development than single measurements [10].

In our cohort, 92% were overweight (BMI $\geq$ 25 kg/m$^2$) and 69% were obese (BMI $\geq$ 30 kg/m$^2$). The extent to which there is a causal connection between BMI and DR and whether being overweight has an influence on the existing DR is a matter of debate. Obesity is an important component of the metabolic syndrome. Klein et al. were able to show in a cohort of type 1 diabetics that an increased BMI is associated with an increased risk of DR development and progression. Every BMI increase of 4 kg/m$^2$ was related to an increase in the risk of DR development by 17% and progression to PDR by 21% [16]. On the other hand, other studies in type 2 diabetics could not show an independent correlation between BMI and DR or even show an inverse correlation with a lower risk of severe DR in overweight type 2 diabetics [20, 26–29]. Not only the BMI, but also the fat distribution could play a role. An abdominal fat distribution (increased waist-to-hip ratio) seems to correlate positively with DR incidence and DR severity [27, 29]. The extent to which a reduction in BMI or, in particular, visceral fat influences the DR incidence or progression has not yet been adequately addressed. There is evidence that bariatric surgery has a positive effect on DR incidence but not on DR progression [30].

In addition to hypertension and obesity, dyslipidemia is one of the most common comorbidities in diabetics [7]. In our study, 93% of the patients presented with elevated LDL cholesterol despite lipid-lowering therapy in 68% at study enrollment. Previous studies have already provided evidence that dyslipidemia appears to influence DR. Cardoso et al. revealed that elevated LDL cholesterol is associated with an increased risk of DR progression in type 2 diabetics [10]. Klein et al. also showed that elevated HDL cholesterol is associated with a lower risk of PDR and DME development [31]. Chung et al. confirmed in a retrospective cohort analysis in type 2 diabetics that elevated HDL cholesterol reduces the risk of DME development [32]. In

addition, there is evidence that the DR and DME incidence is reduced among patients with statin treatment but increased in patients with hypertriglyceridemia [19, 32, 33]. In a large population-based cohort study, patients with statin treatment were significantly less likely to have NPDR, PDR, vitreous hemorrhage, traction retinal detachment, and DME, and were less likely to require intravitreal injections, laser intervention, or vitrectomy than patients without statins [33]. Another established cardiovascular risk factor is Lipoprotein(a). Lp(a) is a lipoprotein which is similar to LDL cholesterol morphologically but contains an additional glycoprotein apolipoprotein(a) [34]. The plasma level of Lp(a) is mainly genetically determined and therefore underlies only a few fluctuations [34]. In our cohort, 28% showed Lp(a) levels above 75 nmol/l, which is considered to be associated with an increased risk for cardiovascular events [34]. To what extent an elevated Lp(a) also has an influence on DR is not yet understood. Funatsu et al. provided evidence that elevated Lp(a) might be an independent risk factor for the progression of NPDR in type 2 diabetics [17]. Kim et al. also showed higher Lp(a) levels in type 2 diabetics with PDR, compared with patients with NPDR or without DR [35]. Other studies, however, could not show a correlation between Lp(a) levels and DR severity [36, 37]. Which pathophysiological mechanisms might play a role in a possible influence of Lp(a) on DR is not understood. An induction of retinal ischemia by the atherogenic effect of Lp(a) with consecutive atherosclerosis and capillary occlusion is suspected [17]. Lp(a) is also considered to have pro-thrombotic properties due to the sequence homology of apolipoprotein(a) and plasminogen [38]. Furthermore, an increased expression of adhesion molecules with consecutive increased adhesion, transendothelial migration, and cytokine production of monocytes in the context of an increased Lp(a) level is discussed, which could promote the progression of a DR [17].

Diabetic nephropathy represents another common microvascular complication of diabetes. In our cohort, 83% of the patients showed a reduced renal function. A reduced renal function seems to be associated with an increased risk of DR progression [39, 40]. Hsieh et al. reported a twofold to threefold increased risk of PDR development within the study period of eight years in patients with a GFR of 30-60ml/min/1.73m$^2$, and a five-fold increased risk in patients with a GFR < 30ml/min/1.73m$^2$ [39]. Rajalakshmi et al. confirmed that an existing diabetic nephropathy is a risk factor for progression to a vision-impairing DR form [40]. In addition, they also found an increasing risk of DR development in patients with decreasing GFR [40]. Elevated serum VEGF levels may contribute to the association between diabetic nephropathy and DR [41]. To what extent an improvement in renal function affects DR is still unclear.

Over half of the patients (57%) showed increased cardiac biomarkers (high-sensitive troponin T and/or NT-ProBNP). Gori et al. described a twofold to threefold increased risk for the development of a cardiovascular disease in diabetics with elevated cardiac biomarkers compared to diabetics with normal values [42]. Additionally, diabetics with an elevated troponin have significantly more comorbidities and an increased risk of mortality [43]. Whether patients with elevated cardiac biomarkers also have an increased risk of DR and DR progression needs to be further investigated.

Limitations of our work are the descriptive design, the limited cohort size and the unicentric design. It should be considered that a DME cohort of a university hospital might not fully represent all DME patients and that our cohort included only Caucasian patients.

In summary, we characterized a DME cohort with respect to additional existing comorbidities. All patients showed at least one additional comorbidity; 86% of the patients even showed ≥ 3 comorbidities. For the most frequent comorbidities, including hypertension, obesity, dyslipidemia, and renal dysfunction, there is sufficient evidence that these have a negative impact on DR. Furthermore, considering the general health in patients with DME beyond the ophthalmologic findings is crucial to prevent further complications of diabetes such as

macrovascular atherosclerosis. Therefore, it is important that the ophthalmologist considers the patient's general health, interacts closely with the general practitioner and internal specialists, and demands early treatment of possible comorbidities.

## Acknowledgments

We thank Julian S. Pottier for the editorial support and linguistic revision of the manuscript.

## Author Contributions

**Conceptualization:** Catharina Busch, Jan Darius Unterlauft, Daniela Vollhardt, Matus Rehak.

**Data curation:** Catharina Busch.

**Formal analysis:** Catharina Busch, Jan Darius Unterlauft.

**Funding acquisition:** Matus Rehak.

**Investigation:** Catharina Busch, Julius L. Katzmann, Claudia Jochmann, Jan Darius Unterlauft, Daniela Vollhardt, Ulrich Laufs.

**Methodology:** Catharina Busch, Julius L. Katzmann, Claudia Jochmann, Jan Darius Unterlauft, Daniela Vollhardt.

**Project administration:** Catharina Busch, Matus Rehak.

**Resources:** Catharina Busch, Claudia Jochmann, Matus Rehak.

**Software:** Catharina Busch.

**Supervision:** Claudia Jochmann, Peter Wiedemann, Ulrich Laufs, Matus Rehak.

**Validation:** Catharina Busch, Matus Rehak.

**Visualization:** Catharina Busch, Matus Rehak.

**Writing – original draft:** Catharina Busch, Julius L. Katzmann, Jan Darius Unterlauft, Peter Wiedemann.

**Writing – review & editing:** Catharina Busch, Julius L. Katzmann, Claudia Jochmann, Jan Darius Unterlauft, Daniela Vollhardt, Peter Wiedemann, Ulrich Laufs, Matus Rehak.

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
