## [Decision Letter · Decision Letter 0]

13 Apr 2021

PONE-D-21-00608

General Health of Patients with Diabetic Macular Edema – The LIPSIA Study

PLOS ONE

Dear Dr. Busch,

Thank you for submitting your manuscript to PLOS ONE. After careful consideration, we feel that it has merit but does not fully meet PLOS ONE’s publication criteria as it currently stands. Therefore, we invite you to submit a revised version of the manuscript that addresses the points raised during the review process.

Please add the references requested by the reviewer.

We look forward to receiving your revised manuscript.

Kind regards,

Alfred S Lewin, Ph.D.

Academic Editor

PLOS ONE

Journal Requirements:

3. In line with PLOS' reporting guidelines (https://journals.plos.org/plosone/s/criteria-for-publication#loc-3), please ensure that you have provided sufficient detail on participant recruitment (i.e., the location from which participants were recruited) in the Methods section.

5. Thank you for stating the following in the Financial Disclosure section:

[Funding: The study was funded by an grant to MR by Novartis Pharma (Grant number: MRTH258A_FVMR002 ).

The funders had no role in study design, data collection and analysis, decision to publish, or preparation of the manuscript.]. 

We note that you received funding from a commercial source: Novartis Pharma

Reviewers' comments:

Reviewer's Responses to Questions

**Comments to the Author**

1. Is the manuscript technically sound, and do the data support the conclusions?

Reviewer #1: Yes

2. Has the statistical analysis been performed appropriately and rigorously? 

Reviewer #1: Yes

3. Have the authors made all data underlying the findings in their manuscript fully available?

Reviewer #1: Yes

4. Is the manuscript presented in an intelligible fashion and written in standard English?

Reviewer #1: Yes

5. Review Comments to the Author

Reviewer #1: Compliments with the Authors the prospective observational study was well conducted the result obtained is important confirms the finding of other studies that finding at least one additional comorbidity have a negative impact on DR and the importance of interdisciplinary treatment of DME and DR

INTRODUCTION

I recommend adding to the references

PAGE 3, LINE 61 .... …. into severe proliferative forms.(3, 4)

Pacella E, La Torre G, Impallara D, Malarska K, Turchetti P, Brillante C, Smaldone G, De Paolis G, Muscella R, Pacella F. Efficacy and safety of the intravitreal treatment of diabetic macular edema with pegaptanib: a 12-month follow-up. Clin Ter. 2013;164(2):e121-6. doi: 10.7417/CT.2013.1543. PMID: 23698213.

and PAGE 3, LINE 68 …. and dyslipidemia is also discussed.(8, 9)

Consiglio di aggiungere alle references

Stefanutti C, Mesce D, Pacella F, Di Giacomo S, Turchetti P, Forastiere M, Trovato Battagliola E, La Torre G, Smaldone G, Pacella E. Optical coherence tomography of retinal and choroidal layers in patients with familial hypercholesterolaemia treated with lipoprotein apheresis. Atheroscler Suppl. 2019 Dec;40:49-54. doi: 10.1016/j.atherosclerosissup.2019.08.031. PMID: 31818450.

6. PLOS authors have the option to publish the peer review history of their article (what does this mean?). If published, this will include your full peer review and any attached files.

Reviewer #1: No

---

## [Author Response · Author response to Decision Letter 0]

11 May 2021

Prof. J. Heber

Editor in Chief

PLOS One

 May 11th 2021

Dear Prof. Heber,

Dear Dr. Lewin,

Dear Reviewers,

We would like to thank you for your valuable comments on our manuscript and increasing the quality of our work.

We addressed every single point accordingly. 

We hope you find our manuscript suitable for publication in PLOS ONE. 

Sincerely,

Catharina Busch, MD, FEBO, PD Dr. med.

University Hospital Leipzig

Department of Ophthalmology

Liebigstr. 10-14

04103 Leipzig

Germany

Tel: +49341/9721650

Fax: +49341/9721659

 

Reviewer #1: Compliments with the Authors the prospective observational study was well conducted the result obtained is important confirms the finding of other studies that finding at least one additional comorbidity have a negative impact on DR and the importance of interdisciplinary treatment of DME and DR

INTRODUCTION

I recommend adding to the references

PAGE 3, LINE 61 .... …. into severe proliferative forms.(3, 4)

Pacella E, La Torre G, Impallara D, Malarska K, Turchetti P, Brillante C, Smaldone G, De Paolis G, Muscella R, Pacella F. Efficacy and safety of the intravitreal treatment of diabetic macular edema with pegaptanib: a 12-month follow-up. Clin Ter. 2013;164(2):e121-6. doi: 10.7417/CT.2013.1543. PMID: 23698213.

and PAGE 3, LINE 68 …. and dyslipidemia is also discussed.(8, 9)

Consiglio di aggiungere alle references

Stefanutti C, Mesce D, Pacella F, Di Giacomo S, Turchetti P, Forastiere M, Trovato Battagliola E, La Torre G, Smaldone G, Pacella E. Optical coherence tomography of retinal and choroidal layers in patients with familial hypercholesterolaemia treated with lipoprotein apheresis. Atheroscler Suppl. 2019 Dec;40:49-54. doi: 10.1016/j.atherosclerosissup.2019.08.031. PMID: 31818450.

Answer: We thank the reviewer for the valuable comments. We added the suggested references accordingly. Changes to the manuscript: p. 3 line 61 and 68.

---

## [Editor Report · Decision Letter 1]

14 May 2021

General Health of Patients with Diabetic Macular Edema – The LIPSIA Study

PONE-D-21-00608R1

Dear Dr. Busch,

We’re pleased to inform you that your manuscript has been judged scientifically suitable for publication and will be formally accepted for publication once it meets all outstanding technical requirements.

Kind regards,

Alfred S Lewin, Ph.D.

Section Editor

PLOS ONE
---

## [Editor Report · Acceptance letter]

3 Jun 2021

PONE-D-21-00608R1 

General Health of Patients with Diabetic Macular Edema – The LIPSIA Study 

Dear Dr. Busch:

I'm pleased to inform you that your manuscript has been deemed suitable for publication in PLOS ONE. Congratulations! Your manuscript is now with our production department. 

Kind regards, 

on behalf of

Dr. Alfred S Lewin 

Section Editor

PLOS ONE